# Traumatic Brain Injury and Secondary Neurodegenerative Disease

**William S. Dodd** [1], **Eric J. Panther** [1], **Kevin Pierre** [1], **Jairo S. Hernandez** [1], **Devan Patel** [2] and **Brandon Lucke-Wold** [1,*]

1   Department of Neurosurgery, College of Medicine, University of Florida, Gainesville, FL 32610, USA
2   Department of Neurosurgery, Jacobs School of Medicine and Biomedical Sciences, University at Buffalo, Buffalo, NY 14203, USA
*   Correspondence: brandon.lucke-wold@neurosurgery.ufl.edu

**Abstract:** Traumatic brain injury (TBI) is a devastating event with severe long-term complications. TBI and its sequelae are one of the leading causes of death and disability in those under 50 years old. The full extent of secondary brain injury is still being intensely investigated; however, it is now clear that neurotrauma can incite chronic neurodegenerative processes. Chronic traumatic encephalopathy, Parkinson's disease, and many other neurodegenerative syndromes have all been associated with a history of traumatic brain injury. The complex nature of these pathologies can make clinical assessment, diagnosis, and treatment challenging. The goal of this review is to provide a concise appraisal of the literature with focus on emerging strategies to improve clinical outcomes. First, we review the pathways involved in the pathogenesis of neurotrauma-related neurodegeneration and discuss the clinical implications of this rapidly evolving field. Next, because clinical evaluation and neuroimaging are essential to the diagnosis and management of neurodegenerative diseases, we analyze the clinical investigations that are transforming these areas of research. Finally, we briefly review some of the preclinical therapies that have shown the most promise in improving outcomes after neurotrauma.

**Keywords:** neurotrauma; secondary mechanisms; neurodegeneration; therapeutics



## 1. Introduction

Neurodegeneration following traumatic brain injury is a complex process that is initiated by several distinct pathways which overwhelm homeostatic stress responses and trigger cellular degeneration and death. Recent studies have demonstrated a progression of neurodegenerative processes months and even years after traumatic brain injury, termed secondary neurodegeneration. Secondary neurodegeneration can manifest in many ways depending on specific etiology and affected neuroanatomy. Chronic traumatic encephalopathy (CTE) is a well-known disease closely associated with repeated traumatic brain injuries (TBI) [1,2]. Parkinson's disease (PD), frontotemporal dementia (FTD), and other neurodegenerative diseases are less common but can also be induced as a consequence of TBI [3–5]. The precise incidence of CTE is hard to quantify due to diagnostic limitations; however, it has gained notoriety due to the prominence of repeated mild TBI in professional sports [1,6]. On the other hand, severe TBI with greater than 1 h loss of consciousness triples the risk of eventually developing PD [3]. Investigation into neurodegenerative disease secondary to TBI is rapidly evolving due to its complex pathophysiology and important public health implications. This review will briefly summarize the current body of knowledge on secondary neurodegeneration, review important imaging modalities related to its diagnosis and management, discuss how the behavioral manifestations of secondary neurodegeneration can aid in diagnosis, and introduce emerging therapeutic targets for the treatment of these diseases.

## 2. Mechanisms of Neurodegeneration after TBI

The inciting mechanisms of secondary neurodegeneration after TBI are an interdependent set of pathological changes initiated by the primary traumatic injury (Figure 1). The temporal evolution of brain injury after TBI is multidimensional and complex but can be conceptualized as overlapping phases. The "acute injury" phase after TBI is characterized by the predominance of mechanical damage resulting from the initial trauma while the "secondary injury" phase is characterized by the delayed emergence of dysregulated metabolism and inflammation pathways [7–9]. The acute phase is generally defined as the first week post-TBI before transitioning into the secondary injury phase that can last months to years [7,10]. Some also advocate for a "subacute" phase as an intermediary that occurs up to 3 months post-TBI [11]. In any case, oxidative stress seems to be a key mediator in the secondary injury phase, as glutamate excitotoxicity, mitochondria dysfunction, and endoplasmic reticulum (ER) stress all contribute to increased reactive oxygen species (ROS) [12–14]. Depolarization of glutaminergic neurons after TBI results in increased calcium ion influx through NMDA and AMPA receptors [15]. Excess intracellular calcium increases mitochondrial ROS production through several mechanisms including activation of $Ca^{2+}$-calmodulin pathways and disruption of the electron transport chain [16]. Endogenous oxidative stress responses are coordinated by the transcription factor Nrf2 [17]. Nrf2 promotes the expression of many cytoprotective proteins including HO-1, NQO-1, and GCLM, among others. These systems can be overwhelmed and become insufficient to prevent ROS-mediated cellular injury [18]. ROS can cause protein damage and misfolding (discussed further below) but may also be especially harmful through lipid peroxidation [19]. Dysregulation of membrane structures such as caveolae in mice is associated with increased markers of neurodegeneration and neuroinflammation [20]. Increased tissue markers of oxidative stress including lipid peroxidation have been observed as far as 12 weeks post-TBI in rats, indicating these pathological mechanisms do not resolve in the acute phase after TBI [21].

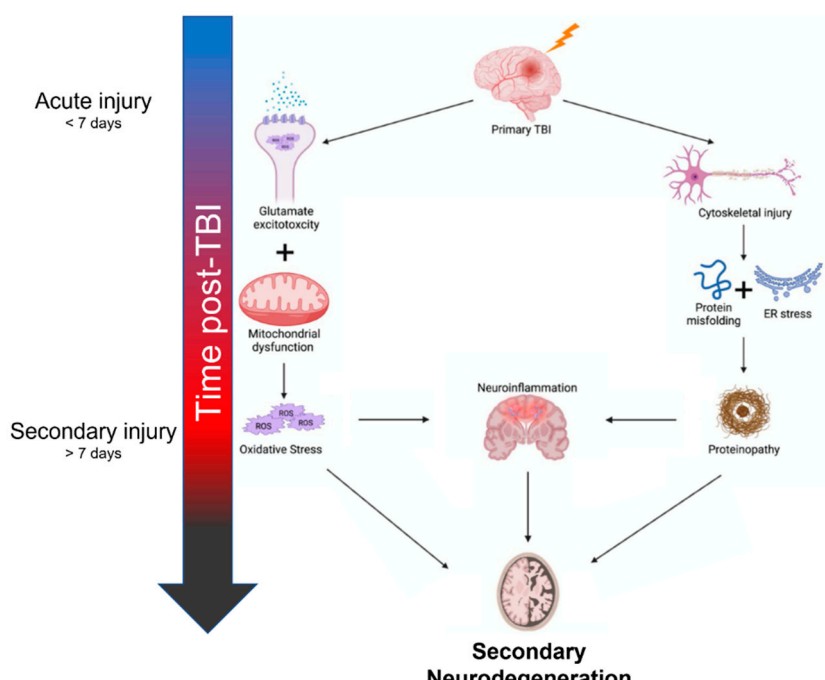

**Figure 1.** Summary of mechanisms contributing to secondary neurodegeneration following traumatic brain injury.

Another central element of secondary neurodegeneration is pathological changes to neuronal cytoskeletal dynamics. Axons, especially within the white matter, are particularly susceptible to damage from tensile strain during traumatic injuries due to their unique

cellular anatomy [22]. The cytoskeleton of these axons can be completely severed during trauma; however, axonal transport may be disrupted even with mild cytoskeletal damage [22]. This type of injury, often termed diffuse axonal injury (DAI) is common after TBI; however, is likely underreported due to the limitations of imaging techniques and the inability to perform brain biopsies in this patient population [23]. Disrupted axonal transport is one of several mechanisms that impede neuronal homeostatic mechanisms and lead to activation of neuroinflammatory pathways (NFκB– and inflammasome-mediated increases in IL-1β, IL-6, TNFα, etc.) and cell death (caspase-3-mediated apoptosis) [24–26]. Pro-inflammatory signals begin locally in damaged neurons but quickly promote reactive gliosis and widespread propagation of the neuroinflammatory cascades by microglia and astrocytes [10,24]. Vascular tissues affected by ROS and pro-inflammatory cytokines are at risk of defective autoregulatory function which can decrease cerebral blood flow and compound cerebral injury [10,25]. These pathological changes result in high protein turnover, particularly in neurons, and may alter ER function by stressing proteostatic mechanisms [27]. ER stress, particularly activation of the unfolded protein response (UPR), is a critical mediator of neurodegenerative change [12,28]. Proteinopathies that occur as a result of misfolding including tauopathies, amyloid plaques, Lewy bodies, and TDP-43 have all been observed after TBI [29–31]. Severe (i.e., associated with >1 h loss of consciousness) TBI triples the risk of developing of Lewy bodies in the substantia nigra [3]. ROS can be both a product and contributing factor to axonal degeneration and neuroinflammation, highlighting the interconnections between mechanisms of secondary neurodegeneration.

## 3. Imaging

Neuroimaging can be used to identify chronic pathological changes from TBI in addition to the acute injury. Generally, TBI disrupts white matter connections and results in cerebral atrophy [32]. This finding tends to be worst in frontotemporal and limbic areas [33], possibly due to trends in traumatic injury mechanisms [34]. Serial quantitative T1 magnetic resonance imaging (MRI) can evaluate neurodegeneration after TBI in a sensitive but non-specific way by assessing cerebral atrophy and volume loss. An increasing ventricle-to-brain ratio is associated with chronic cerebral atrophy in those with TBI after resolution of the acute phase (weeks to months) [35,36]. Generally, TBI-induced neurodegeneration leads to cerebral volumes comparable to that of older individuals with other neurodegenerative diseases; both demonstrating a yearly loss of 1.5% of cerebral volume occurring mostly in sulci and white matter tracts [32,37]. The frontotemporal and limbic areas, which are seated on the sharp sphenoid ridge and edge of the tentorium cerebelli, demonstrate the most severe degenerative changes as their location makes them vulnerable to mechanical deformation. Hippocampal atrophy is especially evident within the limbic system considering its location in the medial temporal lobe and high metabolic demand [38,39]. Patients with DAI-type TBI experience white matter degeneration for months to years following the acute injury as evidenced by studies utilizing MRI diffusor tensor imaging (DTI). DTI is a method for detecting structural changes by analyzing the fractional anisotropy (FA), mean diffusivity (MD), and radial diffusivity (RD) of water molecules. TBI with predominant axonal/white matter injury demonstrates reduced FA and increased MD and RD [40,41]. These findings point to demyelination, loss of axonal integrity, and reduced axonal packing and coherence in frontotemporal and limbic structures such as the anterior limb of the internal capsule, corona radiata, optic radiations, and cingulum [42,43]. Furthermore, changes in these DTI indices are associated with poor neuropsychological performance, including executive function, memory, and functional outcomes [44–47].

Molecular imaging with positron emission tomography is an evolving modality to identify post-traumatic neurodegeneration in vivo in a specific manner. Following TBI, the PET tracer [11]C-Pittsburgh compound-B ([11]C-PiB) binds amyloid-beta (Aβ) in cortical areas, the striatum, and posterior cingulate cortex similar to Alzheimer's disease (AD). Unlike AD, there are increased Aβ depositions in the cerebellum in TBI [48–50]. Additionally, several PET tracers specific to hyperphosphorylated tau (p-tau) previously used in

AD are under investigation for use in TBI. FDDNP is the most well-studied biomarker. FDDNP levels are increased in the midbrain, thalamus, pons, and cingulate gyrus and demonstrate lower binding in temporal and parietal regions in military personnel with mild TBI exposure and football players with suspected chronic traumatic encephalopathy compared to patients with AD [51]; however, FDDNP is non-specific as it also binds Aβ. Other PET imaging biomarkers that bind tau include T801, AV1451, and flortaucipir. Studies are generally limited as they have small sample sizes, lack control groups, or are restricted to one subtype of TBI. While cortical tau tracer uptake varies within individuals with CTE-type TBI, studies have demonstrated consistent uptake in the temporal lobe and limbic system [47,52–54]. The current use of PET imaging biomarkers remains in the early stages. The Enhancing Neuroimaging Genetics through Meta-Analysis (ENIGMA) consortium is evaluating the efficacy of these imaging modalities alone and in combination with fluid biomarkers, radiogenomics, or with EEG. Furthermore, there is evolving research investigating magnetic resonance spectroscopy (MRS), functional MRI (fMRI), transcranial Doppler (TCD), single photon emission computed tomography (SPECT), and functional near-infrared spectroscopy) [53,55,56]. Early human and rodent studies evaluating the newly discovered glymphatic pathway reveal that mild, repetitive TBI alters glymphatic clearance rates examined with MRI [57–60].

## 4. Cognitive and Behavioral Manifestations

Neurotrauma initiates a process of molecular, cellular, and biochemical changes, which subsequently contribute to neuronal damage and death over time. These secondary processes induce damage through apoptosis, inflammation edema, oxidative stress, and mitochondrial dysfunction, which lead to greater damage than the initial insult [61,62]. These downstream effects precipitate long-term consequences, including an increased risk for developing neurodegenerative disorders such as (CTE), Alzheimer disease (AD), unclassified dementia, and Parkinson's disease later in life [63,64].

The clinical management of neurotrauma patients varies widely based on country, hospital structure, comorbidities, and severity, with over 378 practice algorithms identified in a recent review [65]. Appropriate multimodality teams for management of patients with neurotrauma include nutrition, primary care, psychiatry, neurology, neurosurgery, physiotherapy, occupational therapy, speech and language therapy, social services, and neuropsychology [66]. A neuropsychological evaluation consists of a clinical interview to thoroughly evaluate attention, executive function, processing speed, and memory, to capture any current cognitive deficits. The integration of neuropsychology in early treatment can be ascribed to the early recognition of cognitive and behavioral deficits (Table 1), which have been the most devastating chronic problems faced by neurotrauma patients [67,68]. Early treatment of neurodegenerative diseases is crucial for therapeutic success, so prompt detection of these conditions is essential to improve quality of life [69].

**Table 1.** Early behavioral symptoms in different neurotrauma-induced diseases.

| Neurotrauma-Related Disease | Key Behavioral Features | References |
|---|---|---|
| Chronic traumatic encephalopathy | Paranoia, mood swings, apathy, impulsivity, depression, and suicidality | [70–72] |
| Unclassified dementia | Anxiety, apathy, and possibly agitation/disinhibition | [73–76] |
| Parkinson's disease | Motivational decline and slowed thinking | [3,77,78] |
| Alzheimer's disease | Depression, cognitive impairment, memory loss | [79,80] |

In chronic traumatic encephalopathy after mild, repetitive TBI, cognitive findings may precede, follow, or co-occur with behavioral findings. Cognitive symptoms include impaired concentration and attention, disorientation, confusion, and speech abnormalities [70–72]. Behavioral disturbances are often the earliest finding of CTE and can include paranoia, mood swings, apathy, impulsivity, depression, disinhibition, and suicidality [72–74]. An early diagnosis of CTE must involve two or more of the following: pyramidal tract disease, extrapyramidal disease, cognitive and/or behavioral impairment. Neuroimaging studies like PET can aid in the diagnosis [75]. Early signs of unclassified dementia after traumatic brain injury also include apathy, agitation, and disinhibition, but these patients also displayed elevated anxiety, on average, 1.9 years before dementia diagnosis [76].

Parkinsonism is a constellation of symptoms that are characteristically observed in PD. Besides presenting with motor symptoms, PD patients also present with dementia, hallucinations, and cognitive decline. There is considerable evidence suggesting that multiple subtypes of TBI accelerates the neuropathology of PD, therefore early signs of PD should be assessed for early intervention [77]. Behavioral changes are observed early in PD, including motivational decline and slowed thinking [78]. While there is conflicting evidence on epidemiological studies regarding TBI-induced Alzheimer's disease [3], animal models and clinical studies have found a strong link between the two [79]. Depression, cognitive impairment, and memory loss were the first symptoms to appear preceding AD diagnosis [80].

## 5. Therapeutic Targets

Histological analysis and brain imaging studies have revealed many associations between TBI and neurodegeneration. On the other hand, studies on potential therapies to target these pathologies are more limited due to the length of time that often occurs between TBI and the clinical manifestations of neurodegenerative disease. This gap in knowledge is being addressed primarily through animal studies of potential therapeutic interventions. Preclinical therapeutic targets currently being investigated are summarized in Table 2.

### 5.1. Oxidative Stress

Antioxidants and other therapeutics which target ROS generation are appealing translational candidates due to the central role of ROS-mediated cellular damage in the pathogenesis of secondary neurodegeneration. One of the early targeted antioxidant therapies was PEGylated superoxide dismutase (PEG-SOD). The premise of this therapy was that supraphysiological concentrations of SOD would allow rapid detoxification of the ROS generated after TBI. A phase II clinical trial found treatment with PEG-SOD reduced poor outcomes (Glasgow Outcome Scale) at 6 months post-TBI [81]. Many other investigations have turned to naturally-occurring antioxidants. Polyphenols, especially flavonoid-type polyphenols, have shown some promise in reducing oxidative stress after neurotrauma. Several different variations of the flavonoid class compound and some non-flavonoid polyphenols such as resveratrol have been reported in animal studies to reduce tissue oxidation in part through activating the Nrf2 pathway [82–85]. One of the drawbacks of these types of antioxidants is the unclear mechanism of action and bevy of off-target effects. Mitoquinone, a ubiquinone-based molecule modified to preferentially traffic to the inner mitochondrial membrane, solves some of these issues by specifically targeting an organelle known to be dysfunctional after TBI [86]. Mitoquinone treatment in a mouse model of TBI decreased neuronal apoptosis and helped accelerate the antioxidant response of the Nrf2 pathway [87]. Like all preclinical studies, these investigations suffer some limitations. Rodents do not seem to develop neurotrauma-induced neurodegenerative diseases in the same way humans do and studying the long-term consequences of TBI are difficult in short-lived animals.

## 5.2. Cell-Based Therapies

As traumatic brain injury can lead to permanent neurodegeneration and neuronal cell death, there has been significant interest in cell-based therapies to restore neurological function. Cell-based therapies include the use of different types of stem cells including neural stem cells and mesenchymal stem cells [88,89]. Pre-clinical models investigating the utility of neural stem cells have demonstrated improved functional outcomes including improved motor recovery and reduced cognitive deficits [90–92]. In a rodent model of single, severe TBI, treatment with mesenchymal stem cells reduced proinflammatory mediators and increased ant-inflammatory cytokines [93]. Additional studies where mesenchymal stem cells were genetically engineered to overexpress interleukin-10 found enhanced functional recovery after TBI, possibly via alteration of microglial polarization [94].

## 5.3. Aggregation-Prone Proteins

Traumatic brain injuries have been linked to progressive neurodegenerative proteinopathies such as Parkinson's disease and Alzheimer's disease. Thus, pre-clinical models have been used to study the role of aggregation-prone proteins such as Aß and tau proteins and have found that preventing accumulation of these protein pathologies can improve neurocognitive outcomes after mild, repetitive TBI [95–99]. Tau is necessary for normal microtubules function; however, over phosphorylation of tau is associated with proteinopathy development after TBI [100]. Therapies which target phospho-tau protein for degradation are currently being tested with some promising early results [101–103]. Similarly, immunization against pathogenic Aß is an emerging strategy to prevent Aß plaque accumulation [97]. Aß may also be targeted by preventing cleavage of amyloid precursor protein (APP) by amyloidogenic ß- and γ-secretases in favor of cleavage by non-amyloidogenic α-secretase [104–106]. These therapies which enhance clearance or decrease production of Aß could potentially be combined with Aß-binding molecules that disrupt Aß aggregation [107].

## 5.4. Neuroinflammation

Neuroinflammation after traumatic brain injury has been implicated in the subsequent development of neurodegenerative processes [25]. Pre-clinical models have been used to study multiple agents to attenuate this neuroinflammatory response. One group of such agents include pharmacological therapies that are traditionally used for their antimicrobial properties. Minocycline attenuates microglial activation and may reduce secondary brain injury while improving long-term functional outcomes after traumatic brain injury [108–110]. Doxycycline decreases matrix metalloprotease-9 (MMP-9) activity, thus preventing blood-brain-barrier disruption and microvascular hyperpermeability [111]. Hydroxychloroquine and chloroquine have been shown to reduce neuroinflammation by decreasing microglia activation and blood-brain-barrier disruption following cortical impact TBI in animal models [112,113].

Synthetic peroxisome proliferator-activated receptor agonists are another target. In preclinical studies, treatment with PPAR agonists prevented microglial activation and mitochondrial dysfunction after traumatic brain injury and resulted in smaller lesions [114–116]. In a murine model of single, severe traumatic brain injury, treatment with a cannabinoid type 2 receptor agonist decreased neurodegenerative changes. Specifically, treatment attenuated neuron degeneration and blood-brain-barrier permeability and improved behavioral outcomes [117].

In recent years, a growing number of studies have demonstrated the mechanistic role of pyroptosis in neuroinflammatory processes and have identified inflammasomes as a potential therapeutic target [118,119]. In a pre-clinical model of TBI, selective inhibition of the inflammasome resulted in reduced cerebral edema and improved neurological outcomes in association with decreased inflammatory mediators such as caspase-1 and IL-1β [120]. Therapeutic targets for various other downstream neuroinflammatory mediators also have been investigated. Interferon-beta inhibition reduced neuroinflammation, lesion volume,

and long-term neurological impairments in a murine TBI model [121]. Other studies have demonstrated that targeting microglial/macrophage polarization can similarly attenuate neuroinflammation and secondary neurodegenerative processes after TBI [122–125].

**Table 2.** Preclinical Studies of Secondary Neurodegeneration Therapies.

| Therapeutic Target | Outcomes & Mechanism of Action | References |
|---|---|---|
| **Oxidative Stress** | | |
| *PEG-SOD* | Catalyzes the degradation of superoxide radicals; combined with PEG molecules to increase in vivo half-life | [64] |
| *Polyphenols & Flavonoids* | Water-soluble antioxidants; directly react with ROS in addition to stimulating the Nrf2-ARE pathway | [65–68] |
| *Mitoquinone* | Acts as a renewable antioxidant to reduce mitochondrial ROS | [70] |
| **Cell-based Therapies** | | |
| *Neural stem cells* | Improves motor recovery and cognition by replacing neurons lost to neurodegeneration | [73–75] |
| *Mesenchymal stromal stem cells* | Reduces proinflammatory mediators and improved functional recovery. IL-10 overexpression alters microglial polarization in favor of anti-inflammatory processes | [76,77] |
| **Aggregation-prone Proteins** | | |
| *Tau* | Preventing pathologic accumulation via immunization against phosphorylated tau improves neurocognitive outcomes | [84–86] |
| *Amyloid-beta protein* | Aβ may be targeted through enhanced clearance (immunization), decreased production (α-secretase overexpression), or decreased aggregation (Aβ binding molecules). | [87–90] |
| **Neuroinflammation** | | |
| *Minocycline* | Attenuates microglial activation and improves functional outcomes | [92–94] |
| *Doxycycline* | Decreases MMP-9 activity and preserves blood-brain-barrier integrity | [95] |
| *Hydroxychloroquine/chloroquine* | Attenuates microglial activation and preserves blood-brain-barrier integrity | [96,97] |
| *PPAR agonists* | Attenuates microglial activation and mitochondrial dysfunction, decreases TBI lesion sizes | [98–100] |
| *Cannabinoid 2 receptor agonist* | Prevents neuronal degeneration and preserves blood-brain-barrier integrity | [101] |
| *Inflammasomes* | Decreases pro-inflammatory mediators such as caspase-1, IL-18, & IL-1β. Can also reduce pyroptotic cell death by inhibiting gasdermin D cleavage. | [102–104] |
| *Interferon-beta* | Attenuates neuroinflammation, decreases lesion volume, and improves long-term functional outcomes | [105] |

## 6. Conclusions and Future Directions

Development of neurodegenerative disease after traumatic brain injury represents a complex challenge for providing care to patients who have suffered TBI. The multifactorial pathways that contribute to neurodegeneration are difficult to elucidate, but new discoveries are accelerating progress towards effective diagnosis and treatment of these

diseases. Novel imaging technologies and techniques play an important role in this respect, as imaging is critical to initial assessment and long-term tracking of TBI and neurodegeneration. Similarly, understanding how the pathophysiology of neurodegeneration affects neuropsychological pathways is critical to clinical assessment. These investigations also serve the important role of identifying therapeutic targets such as oxidative stress, Proteinopathies, and neuroinflammation. Translational research in this area is crucial to improving long-term outcomes for patients suffering from TBI.

Future inquiry would be most beneficial in addressing the current gaps in knowledge surround TBI and neurodegenerative disease. First, TBI encompasses an enormous array of injury types. There is currently little dedicated literature exploring specific secondary neurodegenerative effects of TBI subtypes such as mild repetitive TBI beyond its association with CTE. Moreover, many studies do not differentiate between specific mechanisms (e.g., crush injury, acceleration/deceleration, penetrating injury, etc.) or anatomical location of primary focal injury. It is reasonable to suspect these variables could impact the secondary neurodegenerative processes that occur after TBI, so future research should explicitly evaluate them.

**Author Contributions:** Conceptualization, W.S.D., E.J.P., K.P., J.S.H., D.P., B.L.-W.; methodology, W.S.D. and B.L.-W.; software, W.S.D., E.J.P., B.L.-W. writing—original draft preparation, W.S.D., E.J.P., K.P., J.S.H., D.P.; writing—review and editing, W.S.D., E.J.P., B.L.-W.; visualization, W.S.D., E.J.P.; supervision, W.S.D. and B.L.-W.; project administration, W.S.D. and B.L.-W. All authors have read and agreed to the published version of the manuscript.

**Funding:** This research received no external funding.

**Institutional Review Board Statement:** Not applicable.

**Informed Consent Statement:** Not applicable.

**Data Availability Statement:** Not applicable.

**Conflicts of Interest:** The authors declare no conflict of interest.

## Abbreviations

CTE—chronic traumatic encephalopathy; TBI—traumatic brain injury; PD—Parkinson's disease; FTD—frontotemporal dementia; AD—Alzheimer's disease; ER—endoplasmic reticulum; ROS—reactive oxygen species; NMDA—N-methyl-D-aspartic acid; AMDA—$\alpha$-amino-3-hydroxy-5-methyl-4-isoxazolepropionic acid; Nrf2—nuclear factor erythroid 2-related factor; HO-1—heme oxygenase 1; NQO-1—NAD(P)H Quinone Dehydrogenase 1; GCLM—glutamate cysteine modulatory gene; DAI—diffuse axonal injury; TDP-43—TAR DNA binding protein 43; MRI—magnetic resonance imaging; DTI—diffusor tensor imaging; FA—fractional anisotropy; MD—mean diffusivity; RD—radial diffusivity; FDDNP—2-(1-6-[(2-[fluorine-18]fluoroethyl)(methyl)amino]-2-naphthylethylidene)malononitrile; TCD—transcranial doppler; SPECT—single photon emission computed tomography; PEG—polyethylene glycol; A$\beta$—amyloid beta; APP—amyloid precursor protein; MMP-9—matrix metalloprotease 9; PPAR—peroxisome proliferator-activated receptor; IL-1$\beta$—interleukin 1 beta.

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
