# Peer review of "Traumatic Brain Injury and Secondary Neurodegenerative Disease"

_traumacare, doi:10.3390/traumacare2040042_

Round 1
Reviewer 2 Report
This review examines mechanisms contributing to neurodegeneration following traumatic brain injury as well as neuroimaging related to the diagnosis of TBI and promising preclinical therapies. Although the review includes useful information, it is overly broad and as a result does not provide a significant contribution to the literature in its present form. Specific suggestions for improvement are discussed below.
Comments:
11. The manuscript would benefit from a clearer title and focus Neurotrauma is an extremely broad term and includes TBI, SCI, stroke, etc. It appears that the authors are largely focused on neurodegeneration following TBI, and further focus on the pathogenesis of TBI-related neurodegeneration. However, even TBI is an extremely broad term and there are multiple types of TBI (focal, diffuse, mild, moderate, severe, etc). This review does not distinguish between mechanisms involved in different types of TBI, which has been highlighted as one of the barriers to finding effective interventions (see J Neurotrauma. 2008 Jul;25(7):719-38. doi: 10.1089/neu.2008.0586.). A focus on either mild TBI/concussion or more severe focal TBI would also be helpful. Finally, it appears that much of the article deals with the links between TBI and neurodegenerative disorders such as AD and PD. This would also be an appropriate focus.
22. The overview of secondary mechanisms involved in TBI is limited to a single paragraph. This is insufficient. For example, neuroinflammation is a major research focus area for TBI and is prominent in Fig 1. Neuroinflammation is discussed under therapeutic targets, but there is very little information on neuroinflammatory mechanisms and the cell types involved. Similarly, calcium dysregulation influences more than oxidative stress. Such a superficial overview is not helpful.
33. For diagnosis, it is surprising that blood and CSF biomarkers are not discussed. The authors have one of the leading experts on TBI, Kevin Wang, as a colleague at the Univ. Florida.
44. The terms primary and secondary are not defined relative to mechanisms.
55. Many statements do not include references.
66. Abbreviations should be defined when first used. For example, it is unclear what FDDNP is the abbreviation for (page 4, line 113)
77. A statement of acknowledgements and/or conflicts of interest (or lack thereof) should be included.
Round 2
Reviewer 1 Report
I feel all concerns have been adequately addressed and that the scope and impact of the review are therefore much improved.
Author Response
Thank you for taking time to review our manuscript.
Reviewer 2 Report
With this revised manuscript, the authors have addressed many of the previous concerns and comments. The focus has been tightened to emphasize the relationship between TBI and late-onset dementing disorders such as Alzheimers disease and Parkinsons Disease. The challenge remains of summarizing a vast amount of information in a relatively short review article. As a result, the review is largely snippets of information for each of the topic areas. Although acknowledged in the final paragraph, it is important to distinguish between various types of TBI (focal, diffuse, mild-severe) for both the implicated secondary mechanisms of neurodegeneration and the potential therapies. Where possible, the manuscript should indicate the relevant type of TBI associated with sentence. For example, DTI imaging is largely related to diffuse/traumatic axonal injury.
Minor comments
1. Citations: it would be helpful to the reader to place the citation after the relevant sentence rather than at the end of the paragraph.
2. Page 2, line 71: New paragraph
3. Page 2, line 78: several mechanisms can lead to inflammation in addition to disrupted axonal transport
2. Page 3, line 105: the Maxwell citation is missing, presumably reference 36
2. Page 4, lines 138 forward: clarify that the cited studies are largely for CTE; lines 148-150 are specific to mild and repetitive mild TBI
Author Response
With this revised manuscript, the authors have addressed many of the previous concerns and comments. The focus has been tightened to emphasize the relationship between TBI and late-onset dementing disorders such as Alzheimers disease and Parkinsons Disease. The challenge remains of summarizing a vast amount of information in a relatively short review article. As a result, the review is largely snippets of information for each of the topic areas. Although acknowledged in the final paragraph, it is important to distinguish between various types of TBI (focal, diffuse, mild-severe) for both the implicated secondary mechanisms of neurodegeneration and the potential therapies. Where possible, the manuscript should indicate the relevant type of TBI associated with sentence. For example, DTI imaging is largely related to diffuse/traumatic axonal injury.
Thank you for bringing up these important points. We agree with the remaining challenges and have made every effort to revise the manuscript to address these points. We have revised the manuscript to be more explicit in describing the type of TBI associated with each discussion point.
Minor comments
- Citations: it would be helpful to the reader to place the citation after the relevant sentence rather than at the end of the paragraph. We have revised the text to ensure citations are after the relevant sentence.
2. Page 2, line 71: New paragraph 3. Page 2, line 78: several mechanisms can lead to inflammation in addition to disrupted axonal transport 4. Page 3, line 105: the Maxwell citation is missing, presumably reference 36 Thank you for catching these errors.
We have revised the text.
2. Page 4, lines 138 forward: clarify that the cited studies are largely for CTE; lines 148-150 are specific to mild and repetitive mild TBI
We have revised the text to clarify types of TBI involved in these discussion points.